# Residual Distillation: Towards Portable Deep Neural Networks without Shortcuts

**Guilin Li**[1*†]**, Junlei Zhang**[1*]**, Yunhe Wang** [1]**, Chuanjian Liu**[1]**,**
**Matthias Tan**[2]**, Yunfeng Lin**[1]**, Wei Zhang**[1]**, Jiashi Feng**[3]**, Tong Zhang**[4]
[1]Noah's Ark Lab, Huawei Technologies [2]CityU, [3]NUS, [4]HKUST
`hiliguilin@gmail.com, zjlnbnb@163.com`
`yunhe.wang@huawei.com, tongzhang@tongzhang-ml.org`

## Abstract

By transferring both features and gradients between different layers, shortcut connections explored by ResNets allow us to effectively train very deep neural networks up to hundreds of layers. However, the additional computation costs induced by those shortcuts are often overlooked. For example, during online inference, the shortcuts in ResNet-50 account for about 40 percent of the entire memory usage on feature maps, because the features in the preceding layers cannot be released until the subsequent calculation is completed. In this work, for the first time, we consider training the CNN models with shortcuts and deploying them without. In particular, we propose a novel joint-training framework to train plain CNN by leveraging the gradients of the ResNet counterpart. During forward step, the feature maps of the early stages of plain CNN are passed through later stages of both itself and the ResNet counterpart to calculate the loss. During backpropagation, gradients calculated from a mixture of these two parts are used to update the plainCNN network to solve the gradient vanishing problem. Extensive experiments on ImageNet/CIFAR10/CIFAR100 demonstrate that the plainCNN network without shortcuts generated by our approach can achieve the same level of accuracy as that of the ResNet baseline while achieving about $1.4\times$ speed-up and $1.25\times$ memory reduction. We also verified the feature transferability of our ImageNet pretrained plain-CNN network by fine-tuning it on MIT 67 and Caltech 101. Our results show that the performance of the plain-CNN is slightly higher than that of its baseline ResNet-50 on these two datasets. The code will be available at https://github.com/leoozy/JointRD_Neurips2020 and the MindSpore code will be available at https://www.mindspore.cn/resources/hub.

## 1 Introduction

Very deep convolutional neural networks (CNNs) have been successfully applied in a large variety of computer vision tasks in recent years [1, 2]. Wherein, the residual modules, i.e., the shortcuts have played a vital role in training very deep neural networks. Shortcuts can be effectively utilized to alleviate the gradient vanishing problem, which is widely used in modern CNN architectures including ResNet [3], MobileNet [4], ResNeXt [5], EfficientNet [6], etc. However, besides the improvement in performance [7], there is an important disadvantage for shortcuts which is often overlooked in existing works. Different from conventional neural architectures (e.g., VGGNet [8]), the feature maps of intermediate layers in those networks using shortcuts cannot be released during online inference. Since the shortcut operation merges features in layers with different depths, we need

---

[*]Co-first authors with equal contributions listed in alphabetical order.
[†]Guilin Li is now affiliated with Intact financial (HK) limited.

to retain their storage for the subsequent calculations. According to Arash et.al [9], for ResNet-152, the shortcuts account for around 43 percent of the total feature map data that consumes much off-chip memory traffic. They also reported a 24.8 percent reduction in energy consumption for ResNet-152 when the shortcut on-chip data is reused.

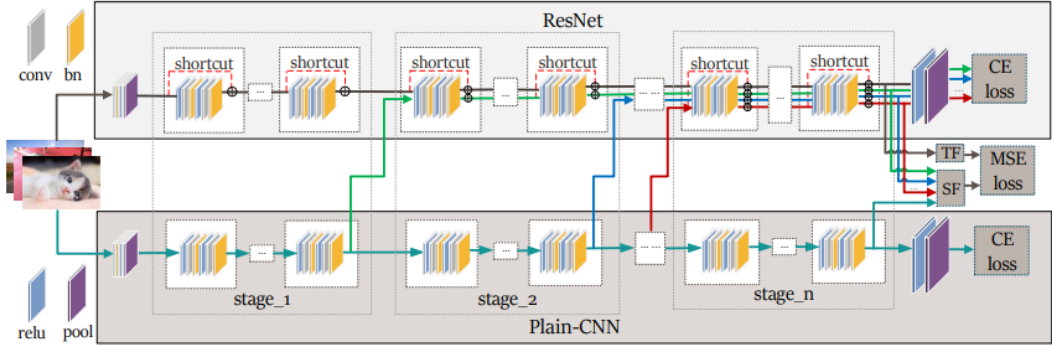

Figure 1: Joint-training framework: early stages of plain-CNN is connected to later stages of ResNet. CE loss represents for the cross-entropy loss, TF represents for teacher feature map and SF represents for students feature maps.

To reduce the redundancy in pre-trained deep networks, a large number of model compression and acceleration approaches have been investigated. Neural Architecture Search [10, 11, 12, 13, 14] searches architectures that consume fewer resources with comparable performance. Model pruning [15, 16, 17] produces models smaller in size via removing the redundant weights or channel. Quantization [18, 19, 20, 21, 22, 23, 24] reduces the precision of the model weights, resulting in smaller model size and faster computation. More deep learning training/inference frameworks [25] are also proposed to optimize the deployment. Although the aforementioned methods have made tremendous efforts for obtaining compact neural networks with reasonable accuracy drop, the potential improvement in deployment efficiency offered by removing the shortcut has been largely ignored. Thus, an algorithm for removing shortcuts without sacrifice accuracy during inference would bring huge benefit. Ideally, this method should also be able to be used on top of other compression methods such as model pruning.

In this paper, we propose to solve the problem mentioned above using the teacher-student paradigm. Different from the current teacher-student framework, our method, for the first time, propose to pass the gradients calculated from the teacher network to the student network. This can also be seen as training the network by adding some auxiliary architecture to assist the convergence. While during inference, the auxiliary part is abandoned. Specially, we develop a Joint-training framework based on Residual Distillation (JointRD), as shown in Figure 1. In practice, the original ResNets are selected as teacher networks, and the students are generated by directly removing shortcuts from the teachers. Then, each of the stages in the plain student network is connected with both later stages of the teacher and student networks. During the back-propagation, gradients are calculated and integrated from both of these two parts. In the early training iterations, gradients from the teacher network play a larger role, which will be gradually reduced until the convergence. By exploiting the proposed joint-training framework, we can effectively integrate the benefit of shortcuts into the training of plain student networks and obtain excellent portable networks without shortcuts.

We verify the effectiveness of the proposed joint-training framework on the ImageNet/CIFAR10/CIFAR100 benchmark datasets. Experimental results show that the student network can achieve comparable accuracy to that of the original ResNet with shortcuts for all these three datasets. For example, the plain ResNet-50 trained using our method achieves a top-1 testing accuracy of 76.08% on ImageNet, which is comparable to that of its baseline teacher network 76.11 %. To check the transferability of the ImageNet pre-trained plain-CNN model using our method, we compare the fine-tuning results of the learned plain-CNN and the ResNet on MIT67 [26] and Caltech101 [27]. Without considering the randomness, the learned plain-CNN outperforms ResNet on these two datasets.

Our method also performs significantly better than traditional teacher-student framework such as logits-based knowledge distillation (KD) ([28])(76.08% verses 71.47%) on ImageNet with ResNet-50. Furthermore, we also valid the effectiveness when using the proposed method together with model pruning ([17]) and KD ([28]). We show that our method can be used on top of either pruning or KD instead of being a substitution of them. Specifically, after removing the shortcut, the network can also benefit from pruning or KD as much as the original ResNet on CIFAR100 and ImageNet.

## 2 Related Works

**Training plain-CNN models:** There have been many efforts made to train plain-CNN models. Zagoruyko et al. [29] proposed to use the Dirac delta initialization to preserve the identity of the inputs in convolutional layers. We adopt this initialization method in our training framework, and in the experiment part, we show that the proposed framework can significantly improve over the naive use of this initialization method. Another attempt is to penalize the contribution of skip-connection [30] and gradually remove the skip-connections during training. However, the accuracy attainable with this method is significantly lower than the ResNets counterpart, which diminishes the benefit of the shortcut removal.

**Knowledge Distillation:** Knowledge distillation (KD) is a method that distils knowledge from strong teacher models to student models. Typically, there are two types of KD frameworks: one proposed by Hinton et.al [28] is to augment the training data by using the predicted logits/soft-label from the teacher model; the other [31, 32, 33] adds the loss (i.e. MSE loss) term to punish the difference between the intermediate features of the teacher models and the student models. Both of these two KD methods try to teach the student what their final learning target is, but in our experiment, we find that it is not always easy for the student to find the proper direction (gradient) by itself to reach this target.

Different from the aforementioned knowledge distillation where losses are imposed to force the student to learn similar outputs like the teacher, our framework allows the student (plain CNN) to use the gradients from the teacher (ResNet). In other words, we not only guide the student by informing them where their destination is but also provide them with step-by-step direction guidance for them to find the way to reach the target. In our results section, we also valid that the proposed methods can significantly outperform traditional KD for this task.

## 3 Residual Distillation for Learning Plain Networks

In this section, we would describe our joint training framework in detail. During training, we train the plain-CNN model with the gradients from both itself and a ResNet model. While during inference, we only use the plain-CNN model and the ResNet part is discarded.

### 3.1 Motivation

One of the motivations for including shortcuts ([3]) in CNNs is to avoid the vanishing gradients problem and to reduce the optimization difficulty. In a recent study, [34] shows that without the use of shortcuts in CNNs, the stochastic gradients produced by different mini-batches sometimes are negatively correlated, which makes it difficult for the optimization algorithm to converge. David Balduzzi [35] also found that the gradients of shallow layers of a 50-layer plain-CNN are similar to white noises and adding the shortcuts can relieve this problem. Inspired by these findings, we argue that plain-CNN models without shortcuts perform poorly because of the poor optimization solution currently adopted instead of the limitations of their expressive power.

One possible solution is to provide the training process better gradients, by utilizing the gradients calculated from the ResNets counterpart. In this work, we develop such a training solution as a Joint-training framework based on Residual Distillation (JointRD). In this framework, the early stages of plain CNN connect to both later

Table 1: Max memory (KB) on the mobile NPU

| Input size | 256 | 384 | 512 |
|---|---|---|---|
| ResNet50 | 438737 | 832733 | OOM |
| plain-CNN 50 | 356657 | 669397 | 1106989 |
| Reduction | 18.7% | 19.6% | - |

stages of itself and later stages of ResNets (as shown in Figure 1).

To verify the potential latency/memory improvement of removing shortcuts, we tested the max memory consumption and latency of plain-CNN 50 and ResNet50 on a mobile NPU. As results shown in Table 1 and Table 2, we could obtain up to 30% speedup on latency and reduce around 19% max memory consumption.

Table 2: Latency (ms) of 50-layer plain-CNN vs ResetNet50 evaluated on the mobile NPU

| input size | 224 | 1024 | 1536 | 2048 |
|---|---|---|---|---|
| ResNet50 | $22.45 \pm 0.34$ | $346.96 \pm 0.64$ | $756.03 \pm 1.94$ | out of memory |
| plain-CNN 50 | $18.03 \pm 0.36$ | $247.65 \pm 1.52$ | $519.01 \pm 3.26$ | $944.59 \pm 4.27$ |
| Speedup | 19.69% | 28.62 % | 31.35 % | - |

## 3.2 The Joint-training Architecture

As shown in Figure 1, the joint-training architecture has two networks: the top one is the ResNet teacher network (network $s$) and the bottom one is the Plain-CNN student network (network $s$). The plain-CNN model is constructed by removing all skip connections from a target ResNet, for example, ResNet50. Therefore, in this training framework, the teacher model and the student model have the same structure except for that the student has no shortcuts. These two models are divided into $N$ stages based on the location of the downsampling layers [3, 36, 37]. For example, a ResNet-50 network [3] is divided into four stages. In the following part, we would take the four-stages ResNet50/Plain-CNN50 as an example to illustrate the ideas.

During the forward step, there would be four forward paths passing through student network $s$, teacher network $t$, and the mixture of them, network $st$. Define $W$ and $V$ as the parameters for the student and teacher network respectively, we have the four paths as:

1. **Path 1:** The input is passed through the first stage of plain-CNN and fed into the second stage of ResNet (the green arrow in Figure 1), following by the third and fourth stages of ResNet, to obtain feature map $f\_st^1(W; V)$ and cross-entropy loss $L\_st^1(W)$.

2. **Path 2:** The input is passed through the first and second stage of plain-CNN and feed into the third stage of ResNet (the blue arrow in Figure 1), following by the fourth stage of ResNet, to obtain feature map $f\_st^2(W; V)$ and cross-entropy loss $L\_st^2(W)$.

3. **Path 3:** The input is passed through the first, second and third stage of plain-CNN and feed into the fourth stage of ResNet (the red arrow in Figure 1), to obtain feature map $f\_st^3(W; V)$ and cross-entropy loss $L\_st^3(W)$.

4. **Path 4:** The input is passed through the whole plain-CNN network (network $s$) until the last stage to obtain feature map $f\_st^4(W)$ and cross entropy loss $L\_s(W)$.

Here, $f\_st(\cdot)$ represents for the student deep nested functions up to the last convolution layer of the last stage, right before they are passed to the relu and pooling layer. Note that although we also adopt regular KD [31, 32] where intermediate features $f\_t(\cdot)$ of teacher models are used to guide the student model in this framework, a key difference is that we also pass the gradients of the teacher to the student.

**Student initialization:** A common belief is that shortcuts in ResNets have two functions, the first is to allow the gradients to flow backward easily and the second is to preserve the features extracted from previous layers. The first function is transferred to plain-CNN by using the joint-training, the initialization method we adopt here handles the second one. ResNets uses an explicit expression to preserve the previous features by using $F(x) + x$. With millions and billions of parameters, it should be possible for the plain-CNN to preserve this in an implicit way such as learning a function such that $G(x) = F(x) + x$. We found that this can be achieved by adopting Dirac delta initialization, proposed by Zagoruyko et al. [29] to preserve the identity of the inputs in convolutional layers.

With the Dirac delta initialization, the weights for the convolution operation can be written as:

$$W_0' = \alpha \cdot W_0 + \beta \cdot I \qquad (1)$$

where $\alpha$ and $\beta$ are learnable parameters, and would be excluded from the weight decay term.

**Teacher initialization:** We use a pre-trained ResNet model to be the initialization of the teacher model and this model will not be updated during the training process.

**Final Layer KD** Although using the KD loss alone would end up with very poor performance on some datasets such as CIFAR100, we found that including it in the joint-training framework would stabilize the training process and improve the final performance of the plain-CNN model. Especially, we minimize the mean square error (MSE) between the final layer output of the teacher model $f_t(\cdot)$ and that of the student models $f_{st}^i(\cdot), i = 1, \cdots, N$, with the feature maps from the student models transformed by a fully-connected regressor :

$$\sum_{i=1}^{N} ||f_t(V), r(f_{st}^i(W; V), w_r)||^2 \tag{2}$$

where $r$ is a fully-connected regressor with parameter $w_r$

### 3.3   Losses and Optimization

To allow the removal of the ResNets from the joint-training framework for efficient deployment, we introduce a temperature weight $\eta$ that controls the contribution of the gradients from the teacher network. At early stages of the training process, the gradients from ResNets play a bigger role with a larger weight, and at the later stages, the gradients contributed by ResNets fade out by using a smaller weight.

We train the parameters $W$ for the student network on condition that the teacher's parameter is known as $V$ using the following loss function $L(\cdot)$:

$$L(W, w_r) = L_s(W) + \eta\{\sum_{i=1}^{N-1} L_{st}^i(W; V) + \lambda \sum_{i=1}^{N} ||f_t(V), r(f_{st}^i(W; V), w_r)||^2\} \tag{3}$$

where $\eta, \lambda$ are the penalty term which determines how much to penalizes the losses from the teacher network. We adopt a cosine annealing policy [38] for $\eta$ to decay from $\eta_{max}$ to $\eta_{min}$.

### 3.4   The Algorithm

---
**Algorithm 1** JointRD Algorithm

---
   **Input:** An initialized JointRD composed of a plain-CNN model $s$ and a corresponding residual model $t$, training set $X$ and the corresponding labels $Y$.
Freeze the weights $V$ of residual model $t$;
**repeat**
   Randomly select a batch $\{x, y\}$
   Feed the data into ResNet $t$ using to get the feature maps $f_t(V; x, y)$
   Feed the same batch of data into **Path 1, 2, 3, 4** in Section 3.2, to obtain feature maps $f_{st}^i(W, V; x, y)$ and losses $L_{st}^i(W), L_s(W), i = 1, \cdots, N$
   Get the overall loss using Equation 3;
   update the weights of the plain-CNN model.
**until** convergence

---

## 4   Experiments

In this section, we evaluate our proposed JointRD on several datasets. First, we verify the effectiveness of our algorithm through experiments with classification datasets: CIFAR-10 [39], CIFAR-100 [40] and ImageNet [41]. To evaluate the transferability of learned features of plain-CNN networks, we further finetune the Imagenet pre-trained plain-CNN models on two downstream task datasets: MIT67 [26] and Caltech101 [27].

We compare our methods with previous knowledge distillation methods [42, 28] and plain-CNN models initialized with the Dirac delta matrix [29] and the related results demonstrate that our method can outperform these methods in most cases.

## 4.1 Experiments on Benchmark Datasets

**CIFAR-10 and CIFAR-100**  We conduct our first set of experiments using CIFAR-10 [39] and CIFAR-100 [40], each containing 50,000 training images and 10,000 test images. CIFAR-10 has 10 classes and CIFAR-10 has 100 classes. ResNet with three different depth are used, i.e., ResNet18, ResNet34, ResNet50, and each of them contains four stages according to the resolution of feature maps.

**Training details**  We train all models 200 epochs, with a learning rate of 0.1, multiplied by 0.1 at epoch 100 and 150. For all models, we set $\lambda = 0.001$ and $\eta$ decreasing with a cosine annealing policy from 1.0 to 0.5 in 60 epochs in the Equation 3. For the initialization of ResNet basic block [3], we direct apply the Dirac delta matrix on each of the convolution operations. While for the ResNet bottleneck block (the input and output dimensions is not the same), we apply the Dirac delta matrix to all convolution operations except for the first pointwise convolution in blocks and the pointwise convolution acting as skip-connections in downsampling layers.

**Results**  We report the mean and standard error of four repeated experiments with four different seeds. In Table 3, we present the results for training plain-CNN without all shortcuts, but having the pointwise convolutions kept for downsampling layers. In Table 4, we present the results of further removing the pointwise convolution at the downsampling layers. In the tables below, "KD (MSE) + Dirac" means that we keep the KD (MSE) losses in the JointRD, but use the cross-entropy loss from the plain-CNN student network. "ResNet" means the corresponding ResNet with the same depth. "Naive" means that we train plain-CNN models naively.

As illustrated from Table 3, our JointRD can train Plain-CNN to achieve comparable accuracy with the ResNet counterpart, significantly outperforming the combination of knowledge distillation and Dirac delta initialization in almost all cases.

Even when we further remove the pointwise convolution in the downsampling layer, we can achieve sufficient high accuracy with marginal accuracy sacrifice.

Table 3: Benchmark results on CIFAR-10/CIFAR-100. Plain-CNN: all shortcuts removed

| Dataset | Model | JointRD (ours) | Naive | KD (MSE) + Dirac | ResNet |
|---------|-------|----------------|-------|------------------|--------|
| CIFAR-100 | plain-CNN 18 | $78.24 \pm 0.04$ | $77.44 \pm 0.10$ | $\mathbf{78.39 \pm 0.10}$ | $77.92 \pm 0.26$ |
|  | plain-CNN 34 | $\mathbf{78.47 \pm 0.22}$ | $72.30 \pm 2.62$ | $77.86 \pm 0.73$ | $78.58 \pm 0.21$ |
|  | plain-CNN 50 | $\mathbf{78.16 \pm 0.20}$ | $55.39 \pm 4.29$ | $50.38 \pm 7.11$ | $78.39 \pm 0.40$ |
| CIFAR-10 | plain-CNN 18 | $\mathbf{95.11 \pm 0.08}$ | $94.81 \pm 0.06$ | $94.72 \pm 0.11$ | $95.19 \pm 0.04$ |
|  | plain-CNN 34 | $\mathbf{94.78 \pm 0.25}$ | $93.73 \pm 0.07$ | $94.50 \pm 0.10$ | $95.39 \pm 0.16$ |
|  | plain-CNN 50 | $\mathbf{94.40 \pm 0.08}$ | $91.13 \pm 0.36$ | $92.27 \pm 0.27$ | $95.31 \pm 0.08$ |

Table 4: Benchmark results on CIFAR-10/CIFAR-100. Plain-CNN$^*$: removed all shortcuts together with pointwise convolutions of downsampling layers

| Dataset | Model | JointRD (ours) | Naive | KD (MSE) + Dirac | ResNet |
|---------|-------|----------------|-------|------------------|--------|
| CIFAR-100 | plain-CNN 18$^*$ | $\mathbf{77.91 \pm 0.21}$ | $76.67 \pm 0.01$ | $77.81 \pm 0.26$ | $77.92 \pm 0.26$ |
|  | plain-CNN 34$^*$ | $\mathbf{78.42 \pm 0.54}$ | $72.72 \pm 0.41$ | $78.41 \pm 0.12$ | $78.58 \pm 0.21$ |
|  | plain-CNN 50$^*$ | $\mathbf{77.68 \pm 0.58}$ | $54.53 \pm 5.57$ | $59.79 \pm 7.68$ | $78.39 \pm 0.40$ |
| CIFAR-10 | plain-CNN 18$^*$ | $\mathbf{95.03 \pm 0.03}$ | $94.78 \pm 0.13$ | $94.67 \pm 0.13$ | $95.19 \pm 0.04$ |
|  | plain-CNN 34$^*$ | $\mathbf{94.62 \pm 0.16}$ | $93.78 \pm 0.13$ | $\mathbf{94.62 \pm 0.03}$ | $95.39 \pm 0.16$ |
|  | plain-CNN 50$^*$ | $\mathbf{94.36 \pm 0.36}$ | $90.59 \pm 0.59$ | $93.11 \pm 0.25$ | $95.31 \pm 0.08$ |

**ImageNet**  The ImageNet dataset consists of 1.2M training images and 50K validation images with 1000 classes. We use our method to train plain-CNN 34 and plain-CNN 50 using this dataset. The results are reported in Table 5.

**Training details** Following [42], we train the whole networks for 120 epochs, with an initial learning rate 0.2, multiplied by 0.1 at epoch 30, 60, 90. The batch size is 512. We set the weight decay as $1e^{-4}$ for resnet34 and Plain-CNN 34. While for the plain-CNN 50, due the removal of shortcuts, the previous weight decay for resnet is too high for plain-CNN 50, therefore we set it as $1e^{-5}$. As for the weight decay for resnet50, we found its accuracy decreases from 76.11% to 75.72% when we decrease its weight decay from $1e^{-4}$ to $1e^{-5}$. So we keep it as $1e^{-4}$ to get a higher baseline. We set the $\lambda = 0.0001$ and $\eta$ decreasing with a cosine annealing policy from 1.0 to 0.5 in 60 epochs in Equation 3. The "Dirac" means that we only apply the Dirac delta matrix and we normalize the weights of convolution layers following [29]. We keep the pointwise convolutions kept for downsampling layers.

From Table 5, plain-CNN models trained with JointRD can achieve comparable results with the corresponding ResNet. Our method also consistently outperforms Dirac delta initialization [29] and KD(MSE) (combined with Dirac delta initialization for a fair comparison [42]), and plain KD(logits).

Table 5: ImageNet Benchmark.

| Model | JointRD (ours) | Dirac | KD (MSE)+Dirac | KD(logits) | Naive | ResNet |
|---|---|---|---|---|---|---|
| plain-CNN 34 | **73.78** | 72.75 | 73.69 | 71.91 | 71.64 | 73.88 |
| plain-CNN 50 | **76.08** | 72.29 | 75.84 | 71.47 | 69.34 | 76.11 |

## 4.2 Transferability of Pretrained Plain-CNN

To examine the transferability of ImageNet pretrained plain-CNN model learned with our method, we fine-tune this pretrained model on two datasets: MIT67 [26] and Caltech101 [27]. Note that the fine-tuning process also requires the using of the JointRD framework. Specifically, we first initialize the plain-CNN student network and the ResNet teacher network with their ImageNet pre-trained weight, then iteratively updating the teacher network and the student network on the downstream target dataset for 100 epochs. The initial learning rate is $1e^{-3}$ and declined to $1e^{-6}$ with a cosine schedule. We set the batch-size as 64 and weight decay as 0.0005. We set the $\lambda = 0.005$ for the MIT67 and $\lambda = 0.5$ for the Caltech101, and we set $\eta$ decreasing from 1.0 to 0.001 in 100 epochs.

| Model | MIT67 | Caltech101 |
|---|---|---|
| ResNet 50 | 80.34 | 96.04 |
| plain-CNN 50* | 81.67 | 96.38 |

Table 6: Transfer learning accuracy

The results presented in Table 6 shows that when considering the extracted feature transferability, plain-CNN surprisingly surpass the teacher model ResNet. This might due to that the JointRD training framework passes the gradients information from both the plain-CNN network and the ResNet network, this can be thought of as assembled gradients, leading to regularized model structure and weight training.

## 4.3 Use JointRD on top of Pruning and KD

In this part, we conduct experiments to show that our method can be used on top of either pruning ([17]) or KD ([28]) instead of being a substitution of them.

In the first experiment, we adopt the pruning method proposed by [17] to prune the ResNet network first with pruning rates of 30% and 60%. Then remove the shortcuts from the pruned network with the proposed method. As shown from Table 7, pruning and the proposed JointRD framework can be used jointly to improve deployment efficiency without affecting each other's performance.

Table 7: Use pruning ([17]) on top of JointRD on CIFAR100.

| Pruning Rate(%) / Model Acc (%) | 0 | 30 | 60 |
|---|---|---|---|
| ResNet-34 Accuracy | 78.42 | 78.52 | 75.71 |
| plainCNN-34 Accuracy | 78.47 | 78.55 | 75.38 |
| ResNet-50 Accuracy | 78.39 | 78.56 | 77.20 |
| plainCNN-50 Accuracy | 78.16 | 78.17 | 77.03 |

In the second experiment, we applied KD proposed by [28] to the training of ResNet and the JointRD framework. As shown in Table 8, our framework can benefit from KD as much as the original ResNet, where the plain-CNN network achieves a slightly higher accuracy than the ResNet counterpart on CIFAR100.

Table 8: Use KD(logits)([28]) on top of JointRD on CIFAR100

| Teacher: | Student | Baseline (%) | KD (%) |
|---|---|---|---|
| ResNet-50: | ResNet-18 | 77.92 | 78.67 |
| | plain-CNN 18 | 77.91 | 79.05 |
| 78.39% | ResNet-34 | 78.58 | 78.98 |
| | plain-CNN 34 | 78.47 | 79.23 |

In the third experiment, we benchmark the benefit brought by the proposed framework comparing with pruning and KD on ImageNet. Although we have shown in previous experiments that our methods can be used on top of pruning and KD, when used stand-alone, our method still outperforms pruning with or without using *KD(logits)*.

## 4.4 Ablation Study

**Contribution of each element in JointRD** we evaluate the performance of simply use the final layer KD or Dirac initialization to verify the effectiveness of the proposed JointRD framework, especially the gradient passing from the teacher network. As we can see from Table 10, the proposed method is significantly better than the single-use of KD or Dirac initialization. This table also confirms the importance or including KD in the JointRD framework to further boost its performance.

**Initialization of ResNet** In the JointRD, the ResNet part is pre-trained and fixed during the training process. Another choice would randomly initialize the ResNet and train it iteratively with the Plain-CNN part. We compare this method with what we used in JointRD, results are provided in Table 11. As we can see that for small datasets such as CIFAR-100, these two methods perform similarly, and the large datasets such as ImageNet, our JointRD performs significantly better. We will report more ablation studies in the supplementary material due to the page limited.

**Effect of of decaying factor of the teacher network** In Equation 3, we reported that a penalty factor to decay the influence of the teacher network gradients. To test the effect of this policy we conducted a comparison experiment on CIFAR100 using plain-CNN 50, the result shows that this decay policy works significantly better than a constant one: 78.16% verses 74.23%.

## 4.5 Visualization of Learned Features and Gradients

To verify whether the proposed JointRD framework drives the plain-CNN part learns features similar to those ResNet extracted, we visualized the intermediate features maps of plain-CNN50 models trained with JointRD on CIFAR10 and their corresponding ResNet50. As shown in Figure 2, plain-CNN models trained with JointRD learn similar features as the ResNet counterpart does. For comparison, we also visualize the intermediate features maps from the same layer of naively trained plain-CNN. Figure 2 shows that without using the JointRD framework, the features learned by plain-CNN are very different from that of ResNet. This verifies the potential mechanism of the JointRD is to guide the Plain-CNN to learn similar features as the teacher ResNet, and it successfully achieves this goal in this experiment.

To further investigate the statement that the JointRD training framework provides better gradients to the plain-CNN model, we also calculate the gradient confusion (lower the better, a measure

Table 9: Compare with smaller network obtained by pruning on ImageNet.

| model | memory(kb) | latency(ms) | Accuracy (%) | KD Accuracy(%) |
|---|---|---|---|---|
| ResNet50 | 242194 | 166.71 | 76.11 | Teacher |
| Prune 40% | 222642 | 152.86 | 74.68 | 75.13 |
| plainCNN50 | **222182** | **137.45** | **76.08** | **76.32** |

Table 10: Ablation study on the CIFAR-100 dataset

| Model | JointRD(ours) | JointRD(without KD) | KD(MSE) | Dirac |
|---|---|---|---|---|
| plain-CNN 18 | **78.24 ± 0.04** | 75.97 ± 0.35 | 75.59 ± 0.25 | 76.44 ± 0.31 |
| plain-CNN 34 | **78.47 ± 0.22** | 74.86 ± 0.28 | 75.24 ± 0.46 | 75.96 ± 0.27 |
| plain-CNN 50 | **78.16 ± 0.20** | 69.67 ± 2.76 | 70.28 ± 2.93 | 71.12 ± 0.25 |

Table 11: Comparision between pretrained ResNet(JointRD) and random initialize it.

| Dataset | Model | JointRD | random initialize |
|---|---|---|---|
| CIFAR-100 | plain-CNN 18 | 78.24 ± 0.04 | 78.24 ± 0.12 |
| | plain-CNN 34 | 78.47 ± 0.22 | 78.10 ± 0.81 |
| | plain-CNN 50 | 78.16 ± 0.20 | 78.22 ± 0.66 |
| ImageNet | plain-CNN 34 | 73.78 | 72.38 |
| | plain-CNN 50 | 76.08 | 67.45 |

for quantifying optimization difficulty) [34] value for each epoch of the training process of both plain-CNN trained with JointRD and naively trained. From results plotted in Figure 3. We can find that the gradients confusion of plain-CNN trained with JointRD is much lower than that of the plain-CNN model trained naively from the beginning until convergence. This verifies our assumption that the joint-training framework can provide the plain-CNN with gradients with better quality and allow the optimization process to converge properly.

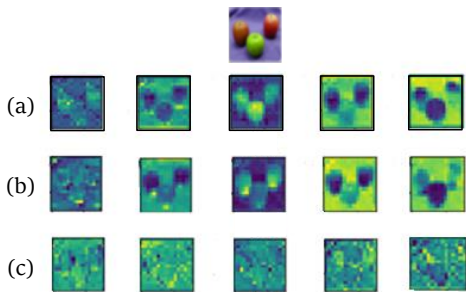 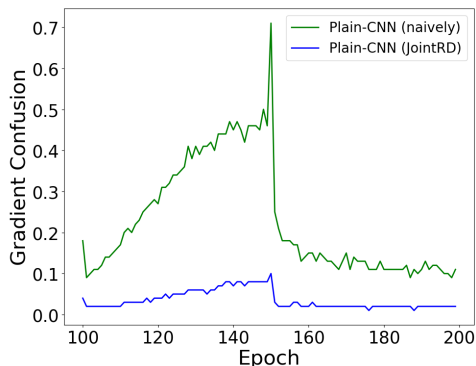

Figure 2: Visualization of intermediate features (a): ResNet (b): plain-CNN trained with JointRD (c): naively trained plain-CNN

Figure 3: Gradient confusion of naively trained plain-CNN50 and JointRD trained one on the ImageNet dataset.

## 5 Conclusion

This work presents a novel approach to deploy portable deep neural networks for mobile devices. There is a great number of model compression and acceleration methods for excavating redundancy in a pre-trained neural network, the online memory usage and computational costs required by shortcuts in modern CNNs have not been fully investigated. We, for the first time, investigated the removal of shortcuts without sacrificing accuracy. Specially, we build a joint-training network such than we can pass the gradient of ResNet to the plain-CNN model to avoid the gradient vanishing problem. Besides experimentally showing that we can remove shortcuts for ResNet models with various depth without accuracy sacrifice, we also verify the generalization ability of this trained plain-CNN network, with the performance of the plain CNN slightly higher than that of its ResNet.

## Broader Impact

The deployment inefficiency caused by shortcuts connections in CNN models has been noticed but largely ignored due to the significant accuracy improvement they bring to CNN models. In this work, for the first time, we consider trains the CNN models with shortcuts and deploy them without. Our experiments show that plain-CNN trained with our method would achieve comparable accuracy with ResNet with the same length while significantly improve the deployment power, memory, and latency efficiency. This work is also among the pioneering works that utilize the gradients of the teacher network to train the student network so that the student network would achieve similar high performance with the teacher network while having favorable deployment property. A potential limitation of this work is that we only try to pass the gradients of a teacher network with the same channel and depth to the student network. An interesting future work would be to explore passing the gradient of a teacher model with a different structure to the student model.

## Acknowledgments and Disclosure of Funding

Funding in direct support of this work: Huawei Noah's Ark Lab.

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
