[Supplementary Material]

# A Comparison with More KD methods

In this part, we attach the performance of another three candidate algorithms to further validate the effectiveness of the proposed JointRD algorithm. The first is to simply use final logits based knowledge distillation proposed by Hinton et.al [22], where soft-label from the teacher model's prediction is used to modify the training label of the student model, we register this method as *KD(logits)*.

The second is to combine the Dirac delta initialization [23] with KD(logits): *KD (logits) + Dirac*. The last one is *KD (MSE_ALL)+ Dirac* in Table 12, where Dirac delta initialization is combined with knowledge distillation from stage-wise intermediate features [35]: the MSE losses between the teacher's intermediate features output and the student's ones for all stages are added in the final loss. Practically, for the final logits based knowledge distillation, we optimize the weights $W_S$ of the plain-CNN model by:

$$\mathcal{L}_{KD}\left(S, T, \tau, y^i, t\right) = \tau \cdot t^2 \cdot \mathcal{L}\left(\sigma\left(\frac{f_S(x^i)}{t}\right), \sigma\left(\frac{f_T\left(x^i\right)}{t}\right)\right) + (1 - \tau) \cdot \mathcal{L}_{CE}\left(\sigma\left(S\right), y^i\right) \quad (4)$$

where $f_S$ and the $f_T$ is the forward function of the student model $S$ and the teacher model $T$ respectively. The $\mathcal{L}_{CE}$ is the cross-entropy loss. The $\tau$ and t are scalar hyperparameters and $\mathcal{L}$ is the Kullback-Leibler divergence loss [22] between the student logits and the teacher logtis. We set $\tau$ as 0.9 and t as 4 following [36]. For the *KD (MSE_ALL)+ Dirac*, the overall loss for the plain-CNN model can be written as:

$$\mathcal{L}_{overall} = \mathcal{L}_{CE} + \alpha\mathcal{L}_{distill} \quad (5)$$

where $\mathcal{L}_{distill}$ is the knowledge distillation loss. Following [35], we set the $\alpha$ as 0.001. Other training parameters are the same with the corresponding JointRD methods.

We conduct repeated experiments with different seeds for the remaining two methods on CIFAR100. The results are presented on Table 12. As we can see from this table, both *KD (logits) + Dirac* and *KD (MSE_ALL)+ Dirac* performs much worse than the proposed methods, suffering from both low accuracy and large variance.

Table 12: Benchmark results on the CIFAR-100 dataset

| Model | JointRD (ours) | KD (logits) + Dirac | KD (MSE_ALL)+ Dirac | ResNet |
|---|---|---|---|---|
| plain-CNN 50 | $78.16 \pm 0.20$ | $70.93 \pm 3.36$ | $63.82 \pm 9.97$ | $78.39 \pm 0.40$ |