[Reviews · NeurIPS 2020]

Review 1

Summary and Contributions: This paper makes the observation that skip connections in ResNet take up nontrivial memory footprint and computational bandwidth. The authors propose to train ConvNets with residual connections, but deploy them without. They show that by jointly training the ResNet and non-ResNet, they are able to achieve good performances on CIFAR and ImageNet classification benchmarks.

Strengths: * This paper has clear figures and explanation of the joint training algorithm. It seems to contain enough details to reproduce the results. * I appreciate the report of runtime statistics on actual mobile platforms (table 1, mobile NPU memory usage, and table 2, mobile inference latency). This is much more useful than proxy metrics like FLOP. * Removing the residual connection at inference time seems to be a novel technique.

Weaknesses: * There are numerous approaches to reduce ConvNet's memory footprint and computational resources at inference time, including but not limited to channel pruning, dynamic computational graph, and model distillation. Why is removing shortcut connection the best way to achieve the same goal? The baselines considered in Table 3 and 4 are rather lacking. For example, how does the proposed method compare to: 1. Pruning method that reduces ResNet-50 channel counts to match the memory footprint and FLOPs of plain-CNN 50. What will be the drop in accuracy? 2. Distill a ResNet-50 model to a smaller ResNet with similar memory footprint as plain-CNN 50. How does this compare to the proposed training scheme? * L107-111 states that "At the early stages of the training process, the gradients from ResNets play a bigger role with a larger weight, and at the later stages, the gradients contributed by ResNets fade out ...". There doesn't seem to be any empirical results or citations to back up this claim. * L133-L143, are there any ablation studies with path 1 - path 3 removed? What is the result of a naive KD method (without those skip connections)? EDIT AFTER REBUTTAL ================== I have read the authors' rebuttal and other reviews. I think the authors have done an excellent job in addressing my concerns. The updated table of results and efficiency metrics are very convincing. Please incorporate this into your final paper! I have thus updated my rating to accept.

Correctness: The results seem correct, but more baselines and ablations are needed.

Clarity: It is fine.

Relation to Prior Work: The discussion is fine.

Reproducibility: Yes

Additional Feedback: * L48 and L96 should cite the same reference - looks like there's a typo. * L92 - L97 is a restatement of L46 - L50. Consider removing to reduce verbosity.


Review 2

Summary and Contributions: The authors proposed a residual distillation approach to train plain CNN using the knowledge of teacher ResNet. The task is well-motivated and the adopted method makes sense. Experiments demonstrate that the performance of the CNN trained this way is on par with the ResNet baseline, but with lower memory and computations.

Strengths: I think overall submission is interesting. The strengths I read are as follows. 1. The experimental results are very promising, on par or even better than the baselines; 2. The shortcut aggregates features and consequently consumes much off-chip memory traffic, which is in fact unfriendly to portable devices that are subject to limited power. This paper introduces a method to distill residuals and to remove shortcuts. Apart from soft logits and intermediate features, the backbone of teacher are used to propagate the features of student, making the optimization easier for plain CNNs.

Weaknesses: 1. The student in this case, is enforced to have the same architecture as the teacher. As a result, a consistency of features between student and teacher is expected. So when would you want to transform the feature maps of Eq.2? 2. The ablation study should be made clearer. Are the Dirac initialization are used in other settings? If no, what is the performance if Dirac initialization is used?

Correctness: Yes to my best knowledge.

Clarity: Yes in general.

Relation to Prior Work: Yes in general.

Reproducibility: Yes

Additional Feedback: Please see the weakness section. ---- Post Rebuttal ---- I have read the rebuttal and keep my original rating.


Review 3

Summary and Contributions: This paper investigates a new training scheme by using which the shortcut of the res-block can be removed without sacrificing much accuracy. The authors propose to use a teacher student training scheme. In addition to use the conventional KD technique, it uses the error gradients from the teacher network to guide the learning of the student network. While this idea is straightforward, the good performance describes in the paper shows that the proposed technique is promising.

Strengths: This paper presents a quite valid motivation of removing the shortcut in resnet for mobile devices. It also quantitatively shows the significant improvement in running speed and memory consumption by using the proposed the method. The strength of the paper is probably coming from the simplicity of the proposed method. With such a straightforward and concise technique the authors show that it is possible to safely remove the shortcut with a minor compromise in accuracy. The proposed method seems novel to me, however I am not an expert in model distillation I am not completely sure about this.

Weaknesses: The authors did not do a good job to use the related work to describe the novelty of the proposed method, hope this can be clarified in the rebuttal. Also, the proposed method seems to have limited advantage in both small models and small tasks. Hope the authors can clarify if this is the case and implication of this if it's true.

Correctness: Should be correct.

Clarity: Yes.

Relation to Prior Work: The authors did not do a good job to use the related work to describe the novelty of the proposed method, hope this can be clarified in the rebuttal.

Reproducibility: Yes

Additional Feedback:

[Author Response · NeurIPS 2020]

The authors sincerely thank all the reviewers for their very constructive and helpful comments.

**Response to Reviewer #1:**

Table 1: Use pruning (Z Liu 2017) on top of JointRD on CIFAR100.

| Model | ResNet-34 | | | plainCNN-34 (JointRD) | | | ResNet-50 | | | plainCNN-50 (JointRD) | | |
|---|---|---|---|---|---|---|---|---|---|---|---|---|
| Pruning Rate (%) | 0 | 30 | 60 | 0 | 30 | 60 | 0 | 30 | 60 | 0 | 30 | 60 |
| Accuracy (%) | 78.42 | 78.52 | 75.71 | 78.47 | 78.55 | 75.38 | 78.39 | 78.56 | 77.20 | 78.16 | 78.17 | 77.03 |

**Q:** *Why JointRD instead of pruning and KD.* **A:** Thanks for your valuable comments. We would like to highlight that our method can be used on top of either pruning or KD instead of being a substitution of them. The results in Table 1 and 2 show that after removing the shortcut, the network can also benefit from pruning or KD as much as the original ResNet.

Table 2: Use KD(logits)(Hinton 2015) on top of JointRD

| DataSet | Teacher: | Student | Baseline (%) | KD (%) |
|---|---|---|---|---|
| CIFAR100 | ResNet-50: | ResNet-18 | 77.92 | 78.67 |
| | | plain-CNN 18 | 77.91 | 79.05 |
| | 78.39% | ResNet-34 | 78.58 | 78.98 |
| | | plain-CNN 34 | 78.47 | 79.23 |

As per your suggestion, we have conducted the filter pruning and KD experiments on ImageNet dataset in Table 3. The performance of plain CNN 50 model trained by the proposed JointRD, is better than the pruned ResNet 50 concerning memory, latency and accuracy, for both cases with or without using KD.

Table 3: Compare with smaller network obtained by pruning and KD: results on ImageNet. Prune: 40% of the channels are pruned.

| model | memory(kb) | latency(ms) | Baseline (%) | KD (%) |
|---|---|---|---|---|
| ResNet50 | 242194 | 166.71 | 76.11 | Teacher |
| Prune 40% | 222642 | 152.86 | 74.68 | 75.13 |
| plainCNN50 | **222182** | **137.45** | **76.08** | **76.32** |

**Q:** *L107-111 gradient.* **A:** In line 171, we reported that a cosine annealing policy is used to decay the penalty factor of the losses from the teacher network. Empirically, this decay policy works better than a constant one: 78.16% verses 74.23% for CIFAR100 on plain-CNN 50. **Q:** *Ablation studies.* **A:** Thanks. For CIFAR100 on plain-CNN 50, the accuracy of original setting, path 1 removed, path 2 removed, and path 3 removed are 78.16%, 75.54%, 76.37%, and 73.74% respectively. **Q:** *Minor comments.* **A:** Thanks. We will correct these typos and proofread the manuscript to make it more readable.

**Response to Reviewer #3:**

**Q:** *Usage of Equation 2.* **A:** In our case where the student has the same number of channels as the teacher, the transformation is used to loosen the constraint of channel-wise Mean-Squared Loss, where only a transformation of the student channel-wise features are required to align with the teacher.

**Q:** *Ablation study.* **A:** Thanks for this nice concern. For the results provided in Table 7, the Dirac initialization are not used, we would refine this table and sentences around it to have an explicit illustration. The performance of using KD together with Dirac is provided in Table 3 and Table 4 instead (the KD(MSE)+Dirac column). As we can see, the proposed method also brings significant benefit over KD(MSE)+Dirac.

**Response to Reviewer #4:**

**Q:** *Comparison to related work.* **A:** Thanks for this constructive comments. The most significant difference of the proposed teacher-student framework from the existing knowledge distillation is the use of the gradients from the teacher models during the training process. Classic knowledge distillation (KD) work either impose loss terms to force the student to learn similar classification soft-label or feature maps like the teacher. Different from the existing methods, our framework allows the student network to use the gradients calculated from the teacher network during optimization. In other words, we not only guide the student by informing them the target points but also provide them with step-by-step direction guidance for them to find the way to the target points. As there are multiple paths in the framework, we also proposed an effective forward and backward process for these paths. These gradients from the teacher model turn out to be very important to achieve good accuracy, compared with only using KD (logits)(Hinton 2015) or KD(MSE)(B Heo 2019): 78.16% verses 70.93% and 63.82% for CIFAR100 on plain-CNN 50 (Table 10). In addition, as shown in Table 1 and 2, our method can be used on top of both pruning and KD(logits). We will include more discussion in the final version.

**Q:** *Small models and small tasks.* **A:** Thanks for this nice concern. The shortcut is explored for avoiding the gradient vanishing for training very deep neural networks on large datasets. For small tasks such as CIFAR100, the accuracy of ResNet-18 and plain CNN18 (naive training) is 77.92% and 77.44%, respectively. Thus, the accuracy improvement using our method is subtle on these shallow models. In contrast, for the ImageNet benchmark, the plain CNN-50 learned using our approach achieves a 76.08% top1-acc with an about 18% latency reduction. We will emphasize this issue and include more discussions in the final version.

[Meta-Review · NeurIPS 2020]

The new training scheme following the teacher-student paradigm to obtain comparable results to those of a resnet model, but without residual connections (shortcuts). Results are on par with SOTA and the approach is very interesting, although not necessarily very novel in principle (I encourage the authors to make this much clearer in the final text). All reviewers agree that this is a good contribution and that the rebuttal was helpful in reaching the final conclusion.